# The Meandrous Route of Rilpivirine in the Search for the Miraculous Drug to Treat HIV Infections

**DOI:** 10.3390/v17070959

**Published:** 2025-07-08

**Authors:** Erik De Clercq

**Affiliations:** Department of Microbiology, Immunology and Transplantation, Rega Institute for Medical Research, KU Leuven, B-3000 Leuven, Belgium; erik.declercq@kuleuven.be

**Keywords:** rilpivirine, cabotegravir, long-acting, intramuscular injection, pain at injection site, people with HIV infection (PWH)

## Abstract

Rilpivirine (RPV, R278474) was highlighted in 2005, two years after the death of Dr. Paul Janssen, as the ideal non-nucleoside reverse transcriptase inhibitor (NNRTI) to treat HIV infections. For this purpose, it was subsequently combined with tenofovir disoproxil fumarate (TDF), tenofovir alafenamide (TAF), darunavir (boosted with ritonavir or cobicistat) or dolutegravir. Its wide-spread use is thanks to its combination with cabotegravir (CAB) in the form of a long-acting intramuscular injection once per month (QM), later twice per month (Q2M), for the treatment of adults, later extended to adolescents and pregnant women, with HIV infections. The long-acting CAB plus RPV should not be administered in patients treated with rifampicin or rifabutin, patients with virological failure or patients with resistance to CAB or RPV, or patients with hepatitis B virus (HBV) infection. Long-acting CAB+RPV may lead to pain at the site of injection which would diminish over time.

## 1. Introduction

To commemorate the untimely death of Dr. Paul A.J. Janssen on 11 November 2003, his colleagues postulated that an ideal anti-HIV drug should (i) be highly active against wild-type and mutant HIV without allowing breakthrough; (ii) have high oral bioavailability and long elimination half-life, allowing once-daily oral treatment at low doses; (iii) have minimal adverse effects; and (iv) be easy to synthesize and formulate [1]. The new diarylpyrimidine (DAPY) non-nucleoside reverse transcriptase inhibitor (NNRTI) R278474 (rilpivirine) was thought to fulfill these criteria. The search for this ideal anti-HIV drug started in 1987, yielding first in 1990 TIBO (tivirapine) [2] and, following a meandrous route, successively α-APA, ITU, DATA, and eventually the DAPYs R147681, R165335 (TMC125), and R278474 (RPV, rilpivirine) [1]. As a closely related analog of rilpivirine, TMC125 was found to be highly effective against wild-type and drug-resistant HIV-1 variants [3,4]. TMC25 displays a high genetic barrier to the development of resistance [5].

## 2. Chemical Structure

The chemical structure of rilpivirine (R278474) is depicted in Figure 1. How it fits within the NNRTI-binding pocket can be seen in Figure 2. How the DAPYs, i.e., TMC125, should be viewed in the scope of other NNRTIs has been further documented by Pauwels [6]. The synthesis of novel DAPY analogs has been optimized by Guillemont et al. [7], and through a series of process optimizations, a practical synthesis method of rilpivirine has been developed, which would be easy to scale up with higher yield and shorter reaction time [8]. The active rilpivirine should be its *E*-isomer, whereas the Z-isomer should be considered as an impurity [9].

## 3. Clinical Use of Rilpivirine

In resource-limited settings (RLS) where integrase inhibitors are not affordable, rilpivirine-based regimens are a good alternative option for HIV-infected individuals who cannot tolerate first-line NNRTI or PI (protease inhibitor) regimens [10].

Rilpivirine is active against wild-type and NNRTI-resistant HIV-1 [11]. Long-acting (LA) injectable RPV has been recommended for pre-exposure prophylaxis (PrEP) against HIV [12,13,14]. As to its metabolism, both the 2-hydroxymethyl derivative and the N-glucuronide of RPV have been identified [15]. Although RPV has been claimed as leading to a low risk of resistance mutations [16], more recent data revealed both E138A and E138G mutations over a longer period and a larger patient population [17]. RPV would not have any adverse effects on pregnancy outcomes [18].

RPV has been postulated to possess activity against acute myeloid leukemia (AML), due to an inhibition of Aurora A kinase [19]. According to Pereira and Vale [20], RPV should have potential for cancer treatment. RPV has also been reported to attenuate liver fibrosis through a selective STAT1-mediated apoptosis in hepatic stellate cells (HSC) [21].

RPV might also offer a possible repurposing in the prevention and/or treatment of Zika virus (ZIKV) replication [22]. It would not interfere with the pharmacokinetics of methadone [23]. However, it may interact with the metabolism of rifapentine [24]; thus, co-administration of rilpivirine and rifapentine should be avoided.

## 4. Combination of Rilpivirine (RPV) with Tenofovir Disoproxil Fumarate (TDF) or Tenofovir Alafenamide (TAF)

Switching to TDF/FTC (emtricitabine)/RPV from TDF/FTC/EFV (efavirenz) was non-inferior in terms of maintaining complete viral suppression at 24 weeks [25,26] (Figure 3). The use of TDF/FTC/RPV for HIV post-exposure prophylaxis has also been recommended by Chauveau et al. [27]. The switch from TDF to TAF further generated weight gain [28]. TAF/FTC/RPV is non-inferior in efficacy but shows less of a decline in bone mineral density and renal function, as compared to TDF/FTC/RPV [29]. A gradually higher prevalence of metabolic syndrome among people living with HIV (PLWH) were noted with changes from RPV/TDF/FTC to RPV/TAF/FTC but plateaued beyond 2 years. However, fewer drugs for dyslipidemia, diabetes, and hypertension were prescribed within the first year after switching to RPV/TAF/FTC [30].

In macaques infected with SHIV (simian/human immunodeficiency virus), oral FTC/TAF and long-acting cabotegravir/rilpivirine achieved an ultra-long-acting antiviral activity that persisted after treatment cessation [31].

## 5. Combinations of Rilpivirine (RPV) with Darunavir (Boosted with Either Ritonavir or Cobicistat) or Dolutegravir

Dual therapy of RPV with boosted-darunavir (bDRV) proved to be effective and safe in patients with advanced HIV infection [32] (darunavir was boosted with either ritonavir or cobicistat). The combination of RPV with darunavir boosted with ritonavir has also shown its effectiveness in patients with a long-lasting HIV infection [33]. The finding that darunavir/ritonavir in combination with RPV was successful in suppressing HIV infection was also found from the 96-week results of Di Cristo et al. [34]. The combination of RPV plus cobicistat-boosted darunavir effected a sustained virological suppression, demonstrated by Maggiolo et al. [35]. The combination of abacavir/lamivudine plus RPV for both the first-line and maintenance therapy of HIV-1 infections was advocated by Ho et al. [36] and Lim et al. [37].

The use of dolutegravir (DTG) (Figure 4) plus RPV as a dual regimen in virologically suppressed HIV-1 infected patients was first mentioned by Casado et al. [38]. DTG plus RPV provided maintenance of virological suppression [39]. The combination of DTG+RPV sustained suppression of HIV-1 associated with a low frequency of virological failure and a favorable safety profile [40]. The combination of DTG plus RPV may move to a first-line therapy in some lower- and middle-income countries [41]. Durable suppression and low rate of virological failure for 3 years have been attributed to the combination of DTG+RPV [42]. Mehta et al. [43] concluded that the DTG+RPV switch was a safe and effective treatment of HIV infections. As a single-tablet regimen (STR), the combination of RPV+DTG would be a cost-effective and long-lasting treatment strategy for PLWH [44,45]. The combination of RPV+DTG would offer some benefits beyond viral suppression, as this combination would slightly improve the immune status during the first 48 weeks after switching, in terms of both CD4+ T-cell and CD8+ T-cell counts, with persistently high rates of viral control [46]. No consistent pattern of change in biomarkers post-switch to RPV+DTG was observed through weeks 48 and 148 in SWORD-1/SWORD-2 while maintaining virologic suppression [47]. Both the combinations DTG+RPV and DTG+3TC (lamivudine) would be efficacious and safe in curtailing virus replication in people living with HIV (PLWH) [48,49]. In recent years (2023–2024), several studies have repeatedly highlighted the effectiveness of RPV+DTG in the treatment of HIV infections [50,51,52].

## 6. Combination of Rilpivirine (RPV) with Cabotegravir (CAB): 2019–2021

Long-acting rilpivirine (RPV) injected intramuscularly once 4- or 8-weekly together with cabotegravir (CAB) (Figure 5) in the treatment of HIV infection was first launched in 2019 [53]. Co-administration of rifampicin (for treatment of tuberculosis) was predicted (based on pharmacokinetic modeling) to result in subtherapeutic concentrations of both RPV and CAB [54]. That notion that monthly injections of long-acting RPV+CAB were non-inferior to standard oral therapy (i.e., dolutegravir-abacavir-lamivudine) was subsequently shown in the ATLAS and FLAIR studies [55,56]. The plasma concentrations of RPV and CAB in the FLAIR study are depicted in Figure 6. The ATLAS and FLAIR trials were hailed as important milestones in the development of HIV therapeutics [57]. The potential of a monthly injectable option for people living with HIV was again emphasized by Murray et al. [58] and Rizzardini et al. [59]. The combination of RPV plus CAB has been approved as Cabenuva™ [60]; it is available at two dosages: CAB (400 mg)+CPV (600 mg) or CAB (600 mg)+RPV (900 mg) [61]. The dual regimen of long-acting RPV and CAB achieved therapeutic concentrations in the cerebrospinal fluid of HIV-infected subjects [62]. Long-acting RPV+CAB has been recommended for use in routine clinical practice; its rate of confirmed virological failure (CVF) has been estimated at about 1% [63]. RPV+CAB is the first complete long-acting injectable regimen for the treatment of HIV-1 infection [64]; it is widely preferred over oral therapy with RPV+CAB [65].

## 7. Combination of Rilpivirine (RPV) with Cabotegravir (CAB): 2021–2023

The 96-week results reaffirm the 48-week results showing long-acting rilpivirine (RPV) and cabotegravir (CAB) to be non-inferior compared with continuing a standard care regimen in adults with HIV-1 for the maintenance of viral suppression (phase 3 FLAIR study) [66]. This was further confirmed with long-acting RPV and CAB dosed every 2 months (ATLAS–2M study) [67]. CAB is mainly metabolized by uridine diphosphate-glucuronosyl transferase (UGT1A1), and RPV is mainly metabolized by cytochrome P450 CYP3A4; therefore, these agents are susceptible to drug–drug interactions (DDIs) [68]. From the FLAIR, ATLAS and subsequent studies, CAB and RPV have emerged as the first long-acting injectable agents for the treatment of HIV infection [69]. Implementation of these data should be further extended to adolescents, pregnant women, and those with barriers to medication adherence [70].

Few confirmed virological failures (CVF) have been observed [71]: the combination of at least two of the following factors, HIV-1 subtype A6/A1, a body mass index (BMI) of ≥30 kg/m^2^, and RPV resistance-associated mutations, was associated with an increased risk of CVF at week 48. The RPV resistance mutations are depicted in Figure 7 [72]. Reduced RPV susceptibility was observed across HIV-1 subtypes B and A1 with resistance-associated mutations K101E or E138K [73]. In another study (in Botswana), the pre-existing RPV-associated mutation E138A was observed [74].

The most common adverse effect in several studies with long-acting RPV+CAB was injection site reactions (ISR, i.e., pain) [75], which improved/resolved with subsequent administrations. How the ISRs evolved over time are depicted in Figure 8 [76]. ISRs should not be ignored as they may limit utility for many patients [77].

The acceptability of different long-acting CAB-RPV dosing schedules should be evaluated in adolescents and perinatally infected patients living with HIV [78]. This evaluation should also be extended to pregnant women [79]. Long-acting RPV and CAB nanosuspensions could be considered for pediatric HIV antiretroviral therapy [80]. Dosages of RPV and CAB should be adjusted (i.e., increased) if combined with rifampicin or rifabutin [81].

Long-acting CAB+RPV has been touted as an amazing treatment strategy for HIV [82]. Its cost-effectiveness has been repeatedly discussed [83,84,85]. Adherence to scheduled dosing visits is strongly recommended to maintain virological suppression with CAB+RPV long-acting injectable therapy and to prevent loss of virological control and possible resistance development [86].

## 8. Combination of Rilpivirine (RPV) with Cabotegravir (CAB): 2024–2025

Those that chose community delivery of the long-acting combination of rilpivirine (RPV) plus cabotegravir (CAB) found it highly acceptable and feasible [87]. In people with HIV (PWH) initiating long-acting injectable CAB+RPV with initial viremia, 48-week viral suppression (<50 copies/mL) was seen in 92% [88]. Real-world data revealed a few virological failures, which were not associated with the acquisition of resistance mutations [89].

Long-acting CAB+RPV was recommended for use in PWH with viremia who were unable to achieve suppression with oral antiretroviral therapy due to suboptimal medication adherence [90]. Long-acting therapy may enhance immune recovery, as attested by an increased CD4^+^/CD8^+^ ratio [91]. At-home administration of long-acting CAB+RPV may be comparably safe, effective, and satisfactory, relative to in-clinic administration [92]. For adult participants it would not matter whether the intramuscular injections are administered in the vastus lateralis (lateral thigh) muscles [93] or deltoid muscles [94].

Real-world data would not demonstrate differences in virological outcomes for individuals with body mass index (BMI) ≥ 30 kg/m^2^ as compared to those with BMI < 30 kg/m^2^ [95]. Long-acting CAB+RPV was efficacious and well tolerated regardless of the baseline BMI category [96].

Long-acting CAB+RPV given every 4 weeks or 8 weeks could also be recommended in virologically suppressed adolescents aged 12 years and older and weighing at least 35 kg [97].

In the US south among reproductive-aged women, nearly two-thirds reported willingness to try long-acting CAB+RPV antiretroviral therapy [98] (Figure 9).

Long-acting CAB plus RPV dosed once every 2 months (Q2M) was successfully implemented across a range of European locations [99]. This regimen may be beneficial for people with HIV (PWH) who are unable to attain viral suppression on oral therapy [100]. Some cases who were eligible for long-acting CAB/RPV experienced virological failure (VF) in this real-world setting [101]. Across 12 UK clinics, VF occurred in 0.7%, and 6% discontinued CAB+RPV [102]. Long-acting CAB/RPV should also be considered in African treatment programs [103], although individuals with hepatitis B virus (HBV) infection will have to be excluded from this wide-spread implementation [104].

The CAB/RPV implementation study in European locations (five European countries), called the CARISEL study, is illustrated in Figure 10 [105]. The long-acting CAB plus RPV yielded high treatment satisfaction, although participants reported moderate injections site pain, which improved with time [106]. CAB+RPV is the first injectable long-acting combination therapy to be licensed for the treatment of HIV-1 infection. Its suitability has been amply demonstrated [107,108]. Its durability is strongly anticipated.

## 9. Conclusions

When rilpivirine (RPV) was initially identified as an ideal anti-HIV compound to be used for the treatment of HIV-1 infections in people living with HIV, it could hardly be predicted that it would find its major indication for the long-acting combination with the integrase inhibitor cabotegravir (CAB). The first question raised is why RPV was almost routinely combined with CAB and not other HIV integrase inhibitors such as dolutegravir (with which it was combined in some trials) or bictegravir (which had never been selected for such combinations). The use of long-acting CAB+RPV has been limited to HIV-1 infections with virally suppressed HIV RNA (<50 copies/mL), patients without the history of virological failure (subsequently to the use of other anti-HIV drugs) or resistance to either CAB or RPV, and, obviously, patients not co-infected with hepatitis B virus (HBV).

## Figures and Tables

**Figure 1 viruses-17-00959-f001:**
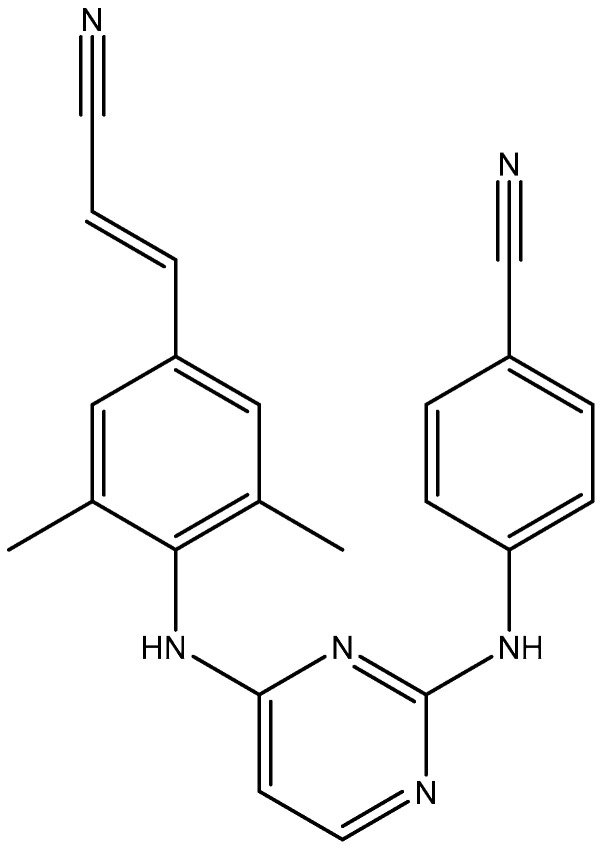
Chemical structure of rilpivirine (R278474).

**Figure 2 viruses-17-00959-f002:**
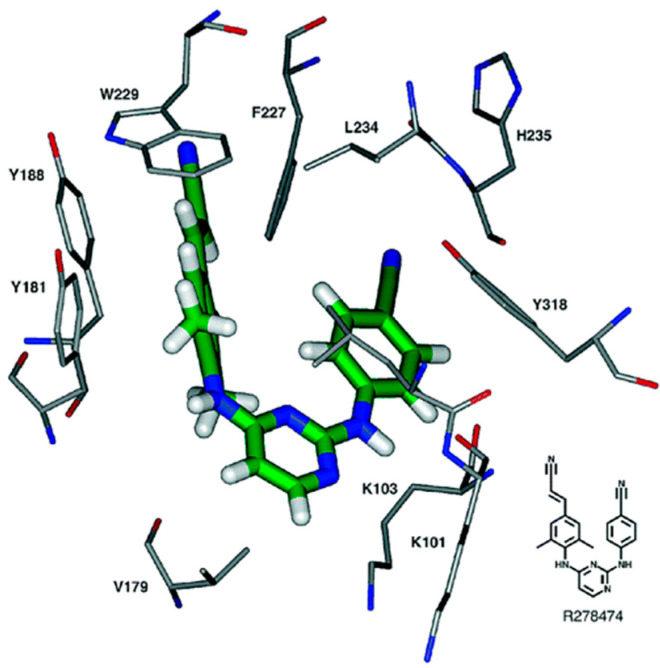
R278474 in the NNRTI-binding pocket (modeled structure) [1].

**Figure 3 viruses-17-00959-f003:**
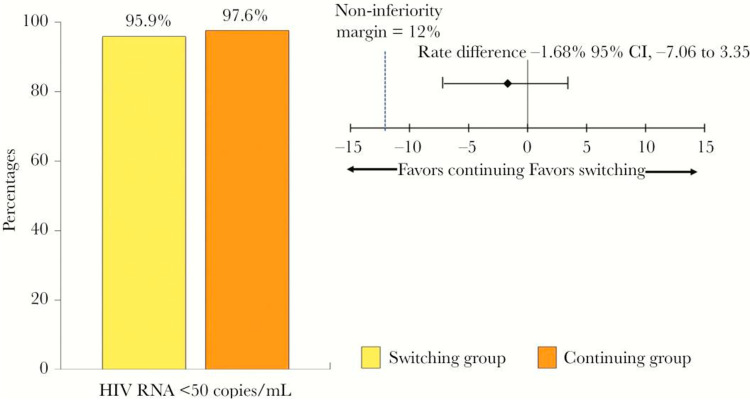
Percentages of patients with HIV RNA <50 copies/mL at week 24 after switching from TDF/FTC/EFV to TDF/FTC/RPV [25].

**Figure 4 viruses-17-00959-f004:**
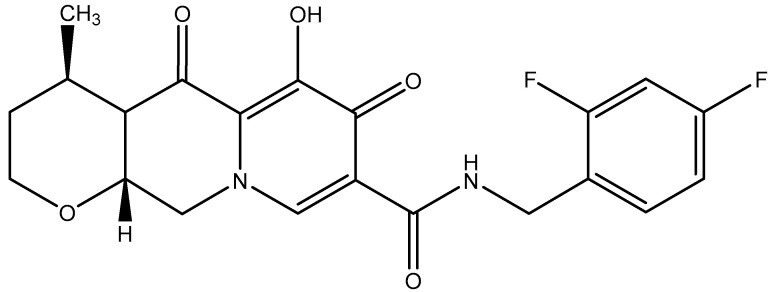
Chemical structure of the HIV integrase inhibitor dolutegravir (DTG).

**Figure 5 viruses-17-00959-f005:**
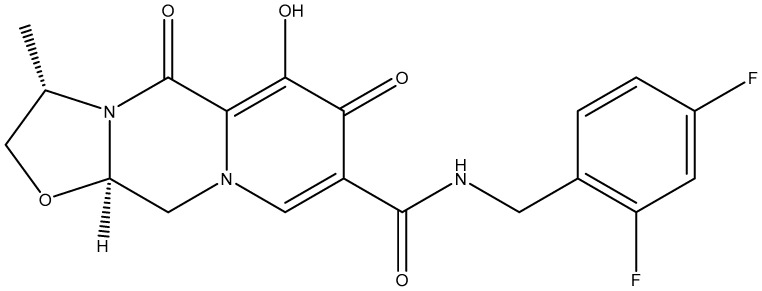
Chemical structures of the HIV integrase inhibitor cabotegravir.

**Figure 6 viruses-17-00959-f006:**
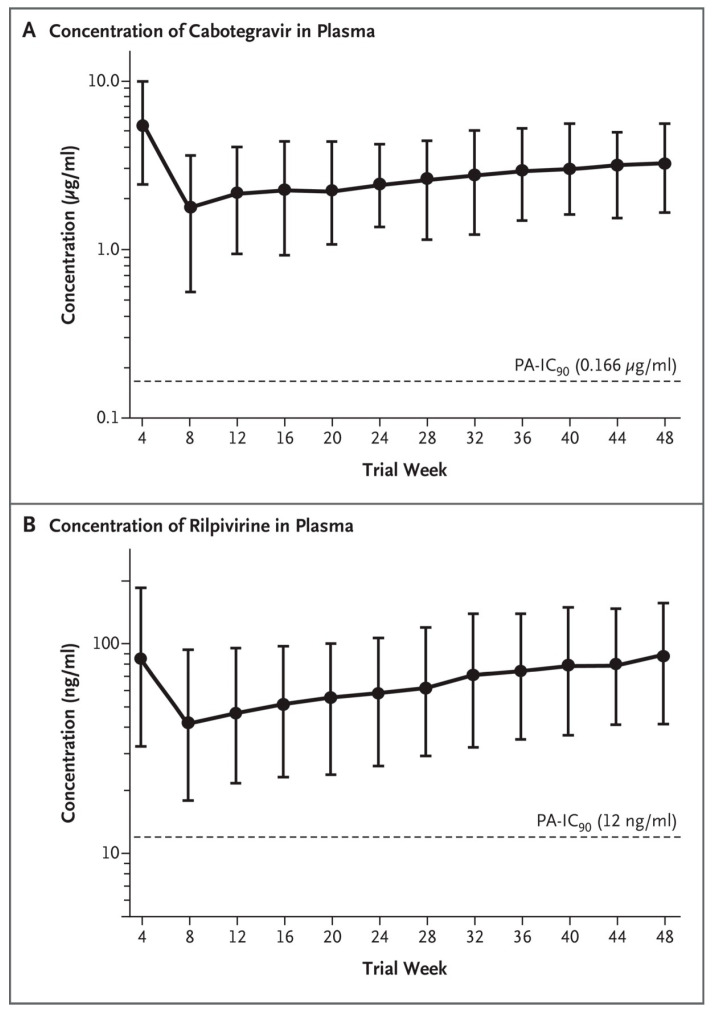
Plasma concentration-time profiles in the FLAIR study [56].

**Figure 7 viruses-17-00959-f007:**
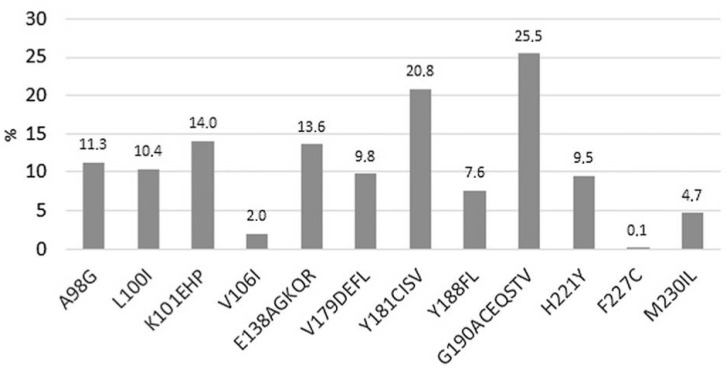
Prevalence of rilpivirine drug resistance mutations in 1372 patients failing NNRTI-based treatment [72].

**Figure 8 viruses-17-00959-f008:**
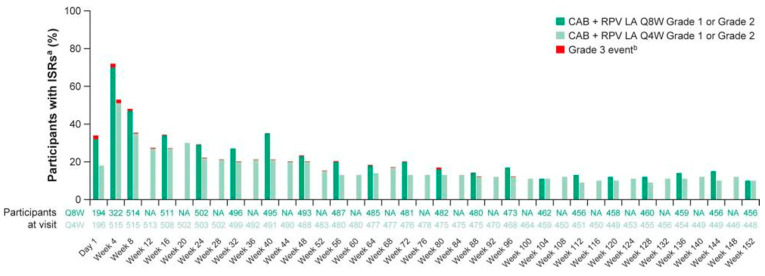
ISRs over time. ^a^ AE grade is the maximum grade reported by the participant at each visit. ^b^ There were no grade 4 or 5 ISRs. Abbreviations: AE, adverse event; CAB, cabotegravir; ISR, injection-site reaction; LA, long-acting; NA, not applicable; Q4W, every 4 weeks; Q8W, every 8 weeks; RPV, rilpivirine [76].

**Figure 9 viruses-17-00959-f009:**
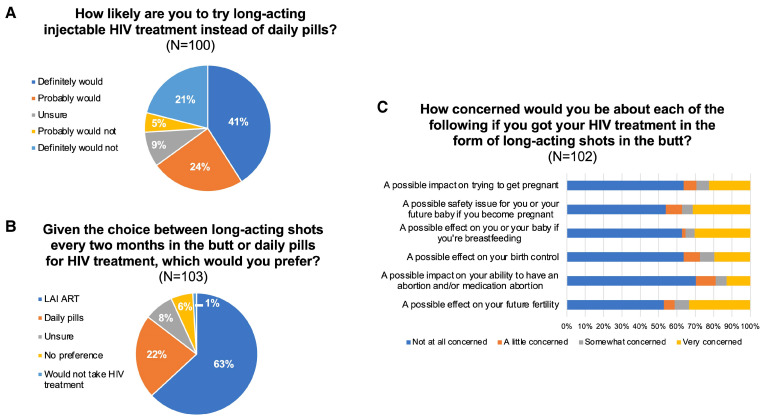
Summary of reproductive-aged women with HIV enrolled in STAR surveyed between December 2022 and October 2023 on willingness to try LAI-ART ((**A**), *n* = 100 had not yet used LAI-ART); preference for LAI over daily pills for HIV treatment ((**B**), *n* = 103); and potential reproductive health concerns related to LAI-ART use ((**C**), *n* = 102, except for 1 participant declined to answer the question related to medication abortion). Abbreviations: ART, antiretroviral therapy; HIV, human immunodeficiency virus; LAI, long-acting injectable; STAR, Study of Treatment and Reproductive Outcomes [98].

**Figure 10 viruses-17-00959-f010:**
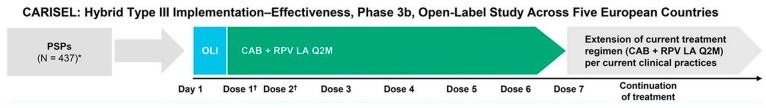
* Four hundred thirty-seven patient study participants (PSPs), and 430 received CAB+RPV LA (long-acting). Oral lead-in (OLI) on day 1. ^†^ Dose 1 was received at month 1, dose 2 at month 2, with the remaining doses Q2M thereafter [105].

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
