# Peer review of "The Meandrous Route of Rilpivirine in the Search for the Miraculous Drug to Treat HIV Infections"

_viruses, 2025, doi:10.3390/v17070959_

Round 1

Reviewer 1 Report

Comments and Suggestions for Authors

The review is dedicated to the history of the development, trials, and current principles of rilpivirine (RPV) use. The author briefly but thoroughly discusses the chemical structure of the drug molecule, its mechanisms of action and resistance as studied in clinical trials. The history of the development of triple and dual regimens including RPV is described. The results of clinical studies on dual regimens in oral form are presented in a concise manner, and, most importantly, the review covers the history and outcomes of trials involving long-acting injectable regimens. The author shares their opinion on these results and outlines the direction for necessary future research.

The issue of resistance was not the focus of this review and, therefore, is not covered in detail. However, if the author could share his perspective on the impact of polymorphic pre-existing mutations to RPV and CAB (L74I and E138A), which are characteristic of certain HIV-1 variants, it could be valuable for many readers. This comment stems from the reviewer’s curiosity and in no way diminishes the value of this review, which is fully ready for publication.

Author Response

Comment 1: "However, if the author could share his perspective on the impact of polymorphic pre-existing mutations to RPV and CAB (L74I and E138A), which are characteristic of certain HIV-1 variants, it could be valuable for many readers."

Response 1: The prevalence of resistance mutations to rilpivirine has already been presented in Fig. 7.

Reviewer 2 Report

Comments and Suggestions for Authors

In this paper, Dr. De Clercq provides a comprehensive review of rilpivirine-based HIV therapy from its historical background to recent advances. This review is generally well-organized. However, some revisions are recommended to improve clarity and consistency.

  1. There are several inconsistencies in the use of abbreviations throughout the paper. Once abbreviation is defined, please use it consistently—for example, rifampicin, rilpivirine, and cabotegravir.
  2. Please provide clear titles for the X-axis in Figure 3 and Figure 7.
  3. Different decimal points are used in Figure 3 and Figure 7. Please standardize them.

Author Response

Comment 1: There are several inconsistencies in the use of abbreviations throughout the paper. Once abbreviation is defined, please use it consistently—for example, rifampicin, rilpivirine, and cabotegravir

Response 1: This requirement has been fulfilled

Comment 2: Please provide clear titles for the X-axis in Figure 3 and Figure 7

Response 2: Both figures have been published previously (Fig. 3 in ref. 25 and Fig. 7 in ref. 72) and cannot be changed anymore

Comment 3: Different decimal points are used in Figure 3 and Figure 7. Please standardize them.

Response 3: Both figures have been published previously (Fig. 3 in ref. 25 and Fig. 7 in ref. 72) and cannot be changed anymore